# Novel Insights into *Phaseolus vulgaris* L. Sprouts: Phytochemical Analysis and Anti-Aging Properties

**DOI:** 10.3390/molecules29133058

**Published:** 2024-06-27

**Authors:** Ewelina Rostkowska, Ewa Poleszak, Agata Przekora, Michał Wójcik, Rafał Typek, Katarzyna Wojciechowska, Katarzyna Dos Santos Szewczyk

**Affiliations:** 1Student Research Group belonging to Chair and Department of Applied Pharmacy, Medical University of Lublin, Chodźki 1, 20-093 Lublin, Poland; ewelinarostkowska@icloud.com; 2Chair and Department of Applied Pharmacy, Medical University of Lublin, Chodźki 1, 20-093 Lublin, Poland; ewa.poleszak@umlub.pl (E.P.); k.wojciechowska@umlub.pl (K.W.); 3Department of Tissue Engineering and Regenerative Medicine, Medical University of Lublin, Chodźki 1 Street, 20-093 Lublin, Poland; agata.przekora-kusmierz@umlub.pl (A.P.); michal.wojcik@umlub.pl (M.W.); 4Department of Chromatography, Institute of Chemical Sciences, Faculty of Chemistry, Maria Curie Sklodowska University in Lublin, 20-031 Lublin, Poland; rafal.typek@poczta.umcs.lublin.pl; 5Department of Pharmaceutical Botany, Medical University of Lublin, Chodźki 1, 20-093 Lublin, Poland

**Keywords:** *Phaseolus*, Fabaceae, sprouts, phenolic compounds, LC-MS, collagen synthesis, antioxidant activity, cytotoxicity

## Abstract

Skin aging is an inevitable and intricate process instigated, among others, by oxidative stress. The search for natural sources that inhibit this mechanism is a promising approach to preventing skin aging. The purpose of our study was to evaluate the composition of phenolic compounds in the micellar extract of *Phaseolus vulgaris* sprouts. The results of a liquid chromatography–mass spectrometry (LC-MS) analysis revealed the presence of thirty-two constituents, including phenolic acids, flavanols, flavan-3-ols, flavanones, isoflavones, and other compounds. Subsequently, the extract was assessed for its antioxidant, anti-inflammatory, anti-collagenase, anti-elastase, anti-tyrosinase, and cytotoxic properties, as well as for the evaluation of collagen synthesis. It was demonstrated that micellar extract from common bean sprouts has strong anti-aging properties. The performed WST-8 (a water-soluble tetrazolium salt) assay revealed that selected concentrations of extract significantly increased proliferation of human dermal fibroblasts compared to the control cells in a dose-dependent manner. A similar tendency was observed with respect to collagen synthesis. Our results suggest that micellar extract from *Phaseolus vulgaris* sprouts can be considered a promising anti-aging compound for applications in cosmetic formulations.

## 1. Introduction

The realm of plants serves as a crucial reservoir of bioactive compounds essential for the development of functional foods, pharmaceuticals, nutraceuticals, cosmetics, and various other applications. Despite the extensive historical use of hundreds of medicinal and aromatic plants in traditional medicine and culinary practices, the efficacy of many species still hinges on anecdotal evidence rather than robust scientific validation facilitated by modern analytical methodologies [1].

Consequently, the utilization of botanical extracts in cosmetic formulations has been on the rise in recent years. Incorporating these extracts into cosmetic products leads to superior outcomes compared to the use of individual bioactive compounds. The quest for natural substances with health-enhancing properties, such as antioxidative, anti-elastase, and anti-collagenase, as well as those devoid of cytotoxic effects on the skin, is experiencing growing demand. Currently, a variety of edible plants are increasingly prevalent in contemporary cosmetic preparations [2]. 

The common bean (*Phaseolus vulgaris* L., F. Fabaceae) is a plant widely distributed throughout the world. It is an herbaceous plant cultivated for its edible beans. In countries with severe malnutrition, its fresh green leaves are also used as food in addition to the beans [3]. The content of active ingredients in green bean leaves was studied by Oyelude et al. [3]. Sterols, tannins, phenolic acids, saponins, and flavonoids were found in the highest amounts in green dried leaves. These components are often used as active ingredients in cosmetic formulations due to their many positive properties [4], such as antioxidant, anti-inflammatory, anti-aging, antiseptic, antifungal, antibacterial, and antiviral [5]. Phenolic acids, including ellagic acid, also exhibit tyrosinase inhibitory activity, thereby preventing excessive melanin formation and skin discoloration [6].

Recently, scientific attention has focused on the beneficial effects of extracts derived from sprouted plants [7]. Sprouts are young shoots that are a few days old and develop from seeds. During germination, the energy reserves contained in the embryo are mobilized, converting fats into free fatty acids, starch into maltose, and proteins into free amino acids. Additionally, vitamins, enzymes, and secondary metabolites (e.g., polyphenols, anthocyanins, and phytosterols) are also synthesized to protect the young shoot from adverse environmental conditions [8]. The use of plant sprout extracts in the cosmetic industry is limited. Fahey et al. [9] were the first to study broccoli (*Brassica oleracea* L.) sprout extracts. They found that the isothiocyanate-sulforaphane content in 3-day-old sprouts is up to 50 times higher than in mature vegetables. This compound exhibits potent antioxidant activity through an activating effect on the transcription factor NF-E2-related factor 2 (Nrf2) [10]. Due to these properties, broccoli sprout extract can be used in anti-aging cosmetics to protect the skin from harmful UV radiation and environmental pollution. Similarly, the extract of 4–5-day-old garden cress sprouts (*Lepidium sativum* L.) can also serve as a valuable cosmetic raw material. This extract is rich in sulforaphane and many nutrients, including vitamins A, C, and E, minerals such as selenium and zinc, and phytonutrients (flavonoids and carotenoids). In cosmetics, these ingredients can have an anti-aging effect in several ways. For instance, sulforaphane, vitamin A, vitamin C, vitamin E, and carotenoids have strong oxidative activity. Furthermore, vitamin A and carotenoids boost collagen production, improving skin elasticity and preventing wrinkles. Zinc and selenium improve skin texture and hydration, helping to maintain skin elasticity [11]. Extracts from 10-day-old sunflower sprouts (*Helianthus annuus* L.) are beneficial due to their high content of vitamins B1, B3, and B6. These vitamins increase cellular energy stored as ATP (adenosine triphosphate), thus extending cell life. The B vitamins in sunflower sprout extract reduce skin dryness and damage, preventing wrinkles [7]. 

Studies indicate that extracts from common beans may have beneficial effects on the skin, including moisturizing, anti-aging, anti-inflammatory, and antioxidant properties [12]. Research conducted by Fonseca-Hernández et al. [13] showed that purified extracts from endemic black beans can be used as an ingredient in cosmeceuticals due to their antioxidant potential and ability to inhibit tyrosinase and elastase enzymes. Similarly, Oh et al. [14] reported that an extract obtained from dried, powdered seeds of *Phaseolus angularis* inhibits aging mediators in human keratinocytes (HaCaT cells) exposed to UVB radiation. Khan et al. [15] obtained a dry extract from the powdered seeds of common beans (*P. vulgaris*), which they then incorporated into a cosmetic cream. Regular use of the cream containing bean seed extract for 8 weeks confirmed its whitening efficacy while reducing erythema, regulating sebum secretion, and moisturizing. Moreover, Lee et al. [16] examined the whitening and anti-inflammatory properties of extracts from beans (*P. radiatus*), showing that they can inhibit melanin production, leading to skin lightening. 

Due to the beneficial effects of extracts obtained from sprouting plants [7], there is an increasing interest in their use. 

In response to the rising demand for high-quality and safe cosmetics, there is a growing interest in products containing plant extracts rich in active compounds. In our study, we aimed to demonstrate the benefits arising from the potential use of edible plants, such as *Phaseolus vulgaris,* in cosmetics. We investigated micellar extract from *P. vulgaris* sprouts to discover new components for anti-aging preparations, evaluating its antioxidant, anti-collagenase, anti-elastase, anti-tyrosinase, anti-inflammatory, and cytotoxic properties, as well as its effects on collagen synthesis. In addition, the qualitative and quantitative composition of the extract was assessed using the liquid chromatography–mass spectrometry (LC-MS) method (Figure 1).

## 2. Results and Discussion

### 2.1. Phytochemical Analysis

The presence of phenolics contributes to the varied colors observed in common bean seeds. Phenolics are present in both the cotyledons and seed coats of common beans, with a greater concentration typically found in the seed coats [17,18]. Numerous studies have shown that sprouts are a rich source of phenolic compounds [19,20]. Polyphenolic compounds have many health benefits, such as antioxidant, antibacterial, and anti-inflammatory effects and protection against cardiovascular diseases [2,21,22]. In the first stage of our work, we assessed the total phenol content [expressed as mg gallic acid equivalent (GAE) per g dry extract], the total phenolic acid content [expressed as mg caffeic acid equivalent (CAE) per g dry extract], as well as the total flavonoid content [expressed as mg quercetin equivalent (QE) per g dry extract]. The obtained results are presented in Table 1. It is noteworthy that the results obtained for *P. vulgaris* sprouts’ micellar extract in our study surpassed those reported for various extracts derived from common beans and sprouts of other species. For instance, Akillioglu and Karakaya [23] found a lower amount of phenolic compounds in a 50% methanol extract of common bean (2.36 mg GAE/g). Hernández-Saavedra and co-authors [24] studied the total phenolic content in the beans of *P. vulgaris*, and they showed that the total phenolic value in the methanol extract was 8.80 mg GAE/g DW. Ombra et al. [25] proved that the acetone extract of common bean whole seeds was in the range of 0.14 to 1.29 mg GAE/g of dry weight (DW), and the extract of the seed coat was in the range of 0.13 to 0.69 mg GAE/g DW. Moreover, the total phenol content for *P. vulgaris* bean water extract was 3.50–5.00 mg GAE/g DW [26]; for 60% methanol with 0.3% HCl, it was 6.66–32.0 mg GAE/g DW [27]; for 40% ethanol with 1% 1 mol/L HCl, it was 11.7–14.7 mg GAE/g DW [28]; and for 70% ethanol (pH = 2), the extract was 1.17–4.40 mg GAE/g DW [29].

Taking into account the total content of polyphenols in the sprouts, the results obtained for micellar extract from *P. vulgaris* sprouts were also higher compared to the results obtained by other authors for various extracts. For example, Guajardo-Flores et al. [30] reported a polyphenol content of 0.90 mg GAE/g DW for an 80% methanol extract of black bean sprouts. *Glycine max* cv. HeiNong and *G. max* cv. Bangladesh soybean-4 sprout extracts contained approximately ~120–150 and ~125–170 mg GAE/100 g FW, respectively [31]; *Vigna radiata* cv. Sulv3 and *V. radiata* cv. BARI mung-4 sprout extracts contained ~75–325 and ~75–320 mg GAE/100 g of fresh weight (FW), respectively [31]; and green mung bean and black mung bean sprout extracts contained 192–1148 and 140–902 mg GAE/100 g DW, respectively [32]. 

The total flavonoid content of the micellar extract of common bean sprouts was estimated using a previously described colorimetric method [33] and expressed as quercetin equivalents (QEs) per gram of dry extracts. The results are presented in Table 1. Similar to the total polyphenol content, the value of the total flavonoid content in the micellar extract was much higher than the values obtained by other authors for various bean extracts. For instance, Aquino-Bolaños et al. [17] demonstrated that the TFC for 70% methanol and 5% acetic acid extracts of common bean seed coat was in the range of 5.90 to 21.5 mg CE (catechin equivalent)/g DW. In the 80% methanol extract of red kidney bean seed coat, Gan et al. [32] found 26.4 mg CE/g DW. For the same kind of extract, these authors reported TFC values of 35.1 mg CE/g DW for big speckled kidney bean seed coat, 35.3 mg CE/g DW for small speckled kidney bean (oval) seed coat, 26.0 mg CE/g DW for violet red kidney bean seed coat, 2.66 mg CE/g DW for brown string bean seed coat, and 44.8 mg CE/g DW for pinto bean seed coat [32]. Our results were also higher than those found by Fernandez-Orozco et al. [34] for germinated mung bean (*Vigna radiata* cv. Emmerald), which contained 1.43–3.46 mg CE/g DW.

The total phenolic acid content (TPAC) in the studied extract is presented in Table 1. The amount found was 4.66 ± 0.29 mg CAE/g DE. Ampofo and Ngadi [35] reported lower values for the total content of phenolic acids. The authors studied the content of phenolic acids in common bean sprouts induced by ultrasound elicitation, ranging from 1.97 ± 1.09 to 216.74 ± 1.26 mg/100 g, depending on the ultrasound parameters and duration.

In the next step of our study, we investigated the chemical composition of the micellar extract obtained from the sprouts of *P. vulgaris* using the LC-MS method. Table 2 shows the identified compounds, including their molecular formula, theoretical and experimental molecular mass, errors in both ppm and mDa, and the fragments. The results of the quantitative analysis are shown in Table 3. The quantities of the compounds identified were determined based on the calibration curves obtained for the standards. For the quantitative analysis of the compounds without available standards, calibration curves for substances of similar structure were used.

In our study, mainly flavonoids and phenolic acids were identified using LC-MS analysis. The chromatogram with marked main peaks is displayed in Figure 2.

A total of 32 polyphenols were identified in the micellar extract of *P. vulgaris* sprouts, comprising 10 phenolic acids (aldaric derivatives of *o*- and *p*-coumaric, ferulic, and sinapic acids), 13 flavanols, 1 flavan-3-ol, 2 flavanons, and 2 isoflavones (Table 3). Among the phenolic acids, caffeic (11,857.2 ± 550.2 μg/g DE) and ferulic acid (11,208.7 ± 558.2 μg/g DE) were identified. The concentration of ferulic acid in our study was significantly higher than the 6.14–18.80 μg/g DM reported by Borges-Martínez et al. [36] for black bean (*Phaseolus vulgaris* L. var. Jamapa) sprouts; the 2640 μg/g of dry matter (DM) reported by Pająk et al. [20] for mung bean (*Vigna radiata* L. Wilczek) sprouts; and the 1870–8260 μg/g DM for green mung bean sprouts and 2100–5570 μg/g DM for black mung bean sprouts reported by Gan et al. [32]. Caffeic acid was previously identified in the sprouts of mung beans [20], but its content was much lower than in the micellar extract. Gan et al. [32] found this acid also in green and black mung bean sprouts. They obtained the highest caffeic acid content in green mung beans after a 5-day germination time. This value was almost four times higher than that obtained for caffeic acid in micellar extract from common beans.

In our study, we also observed significant amounts of *p*-hydroxybenzoic (4693.5 ± 214.0 μg/g DE), *p*-coumaric (1605.7 ± 85.3 μg/g DE), *o*-coumaric (1442.0 ± 65.9 μg/g DE), and sinapic (1442.0 ± 74.6 μg/g DE) acids. Among these, only *p*-coumaric acid had been previously detected at a level of 1.13 ± 0.11 μg/g DE in germinating dark bean (*P. vulgaris* L, c.v. Tolosana) seeds by López et al. [37]. Gallic acid (25.3 ± 1.3 μg/g DE) and vanillic acid (37.2 ± 1.9 μg/g DE) were present in the smallest amounts in the micellar extract, which is consistent with the studies of López et al. [37] and Pająk et al. [20], who also found gallic acid in the sprouts of dark beans and mung beans, respectively, in the smallest amount. The analyzed sprouts contained high amounts of esterified hydroxycinnamic acids, such as *o*- and *p*-coumaric, ferulic, or sinapic acids, which correspond to aldaric acid linked to free acids. *p*-Coumaryl aldaric acid (58,668.2 μg/g DE), *o*-coumaryl aldaric acid (21,923.4 μg/g DE), and feruroyl aldaric acid (10,266.9 μg/g DE) were the most abundant. Aldaric derivatives were previously found in raw, boiled, and germinated beans (*P. vulgaris* L, c.v. Tolosana) [20]. The authors concluded that the aldaric derivatives of ferulic acid (46.14 μg/g DE) were the most abundant in raw beans, followed by derivatives from sinapic (17.34 μg/g DE) and *p*-coumaric acids (6.01 μg/g DE). Interestingly, their concentrations decreased after boiling and germination, with the most significant reduction observed in germinated beans.

Among the identified flavonoids in *P. vulgaris* sprout micellar extract, quercetin derivatives were the most abundant. Rutin, hyperoside, and quercitrin (10,637.5 ± 529.7, 3597.3 ± 173.4, and 2037.9 ± 106.2 μg/g DE, respectively) were found in the largest amount. Rutin was previously identified in germinating dark bean (*P. vulgaris* L, c.v. Tolosana) seeds by López et al. [37], but its content was much smaller (8.55 ± 0.61 μg/g DM). These authors also found hyperoside, but only in raw beans. Interestingly, quercetin-3-*O*-(6″-*O*-malonyl) glucoside and kaempferol-3-*O*-(6″-*O*-malonyl) glucoside, found in quite large amounts in the sprout micellar extract, were previously identified only in dark beans (*Phaseolus vulgaris* L.) [38]. Catechin was determined to be the main flavan-3-ol in *P. vulgaris* sprout extract, similar to the reports of Xu and Chang [39] in raw black beans and Chen et al. [40] in dry bean seed coats.

The findings from the aforementioned authors suggest that germination is a process that alters both the qualitative and quantitative polyphenolic composition of seeds over time, leading to a notable rise in flavonoid and phenolic acid content.

It is worth highlighting that among the compounds identified in *P. vulgaris* sprout extract in the largest amounts are caffeic acid [41], (+)-catechin [42], ferulic acid [43], and rutin [41,44], which ameliorate both intrinsic and extrinsic skin aging. The antioxidant and anti-aging activity of polyphenols is highly dependent on the position, structure, and number of hydroxyl groups. Caffeic acid has two hydroxyl (OH) groups [45], while rutin has five OH groups in the ring, making it easier to oxidize. Rutin and caffeic acid exhibit anti-aging activities through the inhibition of collagenase, tyrosinase, elastase, and hyaluronidase. This finding was proven by Girsang, Ginting, et al. [46], which confirms the potential of rutin and caffeic acid found in snake fruit peels to attach to proteins responsible for the aging processes, namely polyphenol oxidase 3, MMP1, and neutral endopeptidase, demonstrating competitive enzyme inhibition by molecular docking in the skin aging process. Rutin has been reported to inhibit aging-related mitochondrial dysfunction in aged test rats by increasing the oxygen consumption rate, glutathione content, and superoxide dismutase activity, highlighting its antioxidant and anti-aging potential [47]. Caffeic acid is recognized as a highly promising antioxidant, anti-inflammatory, and anti-wrinkle agent [48]. Lee et al. [42] proved that catechin has an antioxidant effect by inhibiting intracellular ROS accumulation in TNF-α-stimulated NHDFs. This compound prevented the degradation of the skin’s extracellular matrix, including the increase in collagenase MMP-1 and the decrease in collagen synthesis. Mechanistically, it acts by suppressing the activation of MAPK, Akt, and COX-2. Furthermore, catechin prevents TNF-α-induced ROS accumulation by capturing free radicals through HO-1. It also suppresses proinflammatory cytokines, including interleukin (IL)-1β and IL-6, which upregulate inflammatory reactions and promote aging-related changes, including skin aging. 

### 2.2. Antioxidant Activity

Because the skin is directly exposed to the environment, it is particularly susceptible to rapid aging due to environmental damage. The primary contributors to extrinsic skin aging are secondary reactions mediated by reactive oxygen species (ROS) that arise when the skin absorbs UV rays [42,49]. Elevated levels of ROS lead to the formation of wrinkles through the cleavage and abnormal cross-linking of fibrous proteins like collagen and elastic fibers in the skin’s extracellular matrix [42,50]. Consequently, in order to slow down skin aging, it is crucial to search for antioxidants that can inhibit the production of ROS. Although numerous synthetic antioxidants are available, they frequently pose potential adverse effects. Therefore, the pursuit of naturally derived antioxidants from plants is of significant importance [51]. 

In our study, we evaluated the free radical scavenging effects of common bean sprout micellar extract using the 2,2-diphenyl-1-picrylhydrazyl (DPPH^•^) and 2,2′-azinobis-(3-ethylbenzothiazoline-6-sulfonate) (ABTS^•+^) assays. The assessment of antioxidant activities was conducted on a microplate scale using cell-free systems and was tested for different concentrations of the extract. The results showed antioxidant activities of 246.24 mg Trolox/g DE in the DPPH^•^ assay and 74.38 mg Trolox/g DE in the ABTS^•+^ assay (Table 4).

Comparing the results from different reports can be challenging due to variations in the experimental conditions. Additionally, the existing knowledge regarding the antioxidant activity of common bean sprouts is incomplete and fragmented. However, Pająk and co-authors [20] found that the methanol extract of mung bean (*Vigna radiata* L. Wilczek) sprouts has an antioxidant activity of 1.41 ± 0.11 mg Trolox/g DM in the DPPH^•^ assay and 11.33 ± 0.34 mg Trolox/g DM in the ABTS^•+^ test. The results obtained by these authors were much higher compared to the results for mung bean seeds, which were 0.11 ± 0.00 and 0.86 ± 0.02 mg Trolox/g DM, respectively.

### 2.3. Anti-Inflammatory Activity

In our study, we examined the inhibitory activities of micellar extract from *P. vulgaris* sprouts on cyclooxygenase-1 (COX-1) and cyclooxygenase-2 (COX-2) enzymes as part of their anti-inflammatory mechanism. COX facilitates the conversion of arachidonic acid into prostaglandins, which play a significant role in both health and various diseases. COX exists in two isoforms: constitutive COX-1, responsible for maintaining normal physiological function, and inducible COX-2, whose expression is activated during inflammatory conditions. Inhibiting COX-1 can lead to certain side effects, whereas inhibiting COX-2 offers therapeutic benefits in treating inflammation, pain, and numerous other diseases [52].

In this study, the micellar extract of P. vulgaris sprouts was tested at different concentrations to assess its inhibitory effects on both cyclooxygenases. The results, expressed as the percentage inhibition of prostaglandin biosynthesis, are shown in Table 5.

The most active concentrations against COX-1 and COX-2 were 50 µg/mL and 100 µg/mL. The anti-inflammatory activity of these extracts is comparable to the inhibition values of the standard compound, indomethacin, at a concentration of 1.25 μM. The micellar extract showed an IC_50_ of 72.43 ± 0.13 μg/mL for COX-1 and 86.37 ± 0.36 μg/mL for COX-2. These results are less potent than those reported by Oomah et al. [53], who found that the acetone extract of black bean hull exhibited strong COX-1 (IC_50_ = 1.2 μg/mL) and COX-2 (IC_50_ = 38 μg/mL) inhibitory effects. Other in vitro findings suggested that alcalase hydrolysates of the common bean pinto Durango inhibited inflammation, with IC_50_ values of 34.9, 13.9, 5.0, and 3.7 μM for the inhibition of cyclooxygenase-2 expression, prostaglandin E2 production, inducible nitric oxide synthase expression, and nitric oxide production, respectively. Moreover, the hydrolysate significantly inhibited the transactivation of NF-κB and the nuclear translocation of NF-κB p65 [54]. 

In our study, an LC-MS analysis of the micellar extract of *P. vulgaris* sprouts revealed the presence of gallic acid, quercetin and its derivatives, and (+)-catechin, which have proven their clinical potential as anti-inflammatory compounds [55,56]. 

### 2.4. Anti-Collagenase and Anti-Elastase Activities

Collagen, a key component of the typical human dermis, primarily upholds its structural integrity. Collagenases, enzymes that break down interstitial collagens, initiate the process of collagen reduction [57]. Inhibiting these enzymes can mitigate collagen degradation and, consequently, postpone the formation of wrinkles in aging skin. The anti-collagenase potential of an extract derived from the sprouts of *P. vulgaris* was assessed using *Clostridium histolyticum* collagenase (Sigma-Aldrich, Steinheim, Germany). The results are presented in Table 6. The highest activity was observed for the common bean sprout extract at a concentration of 50 µg/mL (61.27 ± 0.15%). The IC_50_ value was 59.99 ± 0.10 μg/mL.

In typical adult skin, elastin predominates and plays a crucial role in maintaining the skin’s elasticity [57]. Given the numerous reports indicating a direct correlation between skin aging and the enzymatic breakdown of elastin by elastase [58], in our study, the elastase inhibitory activity of the *P. vulgaris* sprout extract was also assessed. The evaluation was conducted using N-succinyl-L-alanyl-L-alanyl-L-alanine 4-nitroanilide (N-succinyl-Ala-Ala-Ala-*p*-nitroanilide) as the substrate molecule and elastase sourced from porcine pancreas as the enzyme. Epigallocatechin gallate (EGCG) served as the positive control, exhibiting a 91.35% inhibition of enzyme activity. The highest activity was observed for the common bean sprout extract at a concentration of 100 µg/mL (86.29 ± 0.35%). The IC_50_ value was 44.39 ± 0.21 μg/mL.

To the best of our knowledge, previous studies have not reported the in vitro anti-collagenase and anti-elastase activities of *P. vulgaris* sprouts. Although the anti-aging potential of polyphenols from a black bean endemic variety was investigated [13], the authors found that the purified and crude extracts recovered by LC showed a higher capacity to inhibit elastase activity than supercritical fluid extraction (SFE). The SFE-purified extract showed an IC_50_ of 0.023 mg/mL, and for the purified LC extract, the IC_50_ was 0.005 mg/mL. For the crude extracts, SFE presented a value of 0.142 mg/mL, and the LC extract had a value of 0.105 mg/mL.

### 2.5. Anti-Tyrosinase Activity

Tyrosinase is an enzyme that limits the rate of melanin production. The anti-tyrosinase effect was determined by measuring the oxidation of 3,4-dihydroxy-L-phenylalanine (L-DOPA) to dopaquinone using an in vitro fungal tyrosinase assay. The *P. vulgaris* sprout extract at a concentration of 100 μg/mL showed the strongest inhibition (58.33%), and the activity was similar to that of the reference substance, kojic acid (65.49%) (Table 6). The IC_50_ value was 68.80 ± 0.46 μg/mL. The potent anti-tyrosinase activity of the extract can be attributed to the high total content of polyphenols (192.85 mg GAE/g DE), which not only have good affinity for tyrosinase, preventing the formation of dopaquinone, but also bind to it, reducing its catalytic activity [59]. The anti-tyrosinase properties were previously studied by Fonseca-Hernández and co-authors [13] for the black bean endemic variety. They found that the purified extracts obtained by SFE and LC present IC_50_ values for tyrosinase of 0.147 mg/mL and 0.143 mg/mL, respectively. In the raw extracts, the values were 9.92 mg/mL for SFE and 2.59 mg/mL for LC.

### 2.6. Cytotoxic Activity, Cell Proliferation Assessment, and Evaluation of Collagen Synthesis

The cytotoxicity screening test revealed that the common bean sprout extract did not adversely affect cell viability at concentrations as low as 5 and 10 μg/mL (Figure 3). However, when the *P. vulgaris* sprout extract was tested at concentrations ranging from 20 to 160 μg/mL, there was a slight but statistically significant decrease in cell viability to 82–93%, depending on the exposure time and applied extract concentration. Importantly, cell viability remained above 80% and did not decrease with increasing incubation time up to 48 h (Figure 3B). At higher concentrations, notably 320 μg/mL, the extract exhibited a significant reduction in cell viability compared to the control. Specifically, after 24 and 48 h exposure time, cell viability was approximately 75%, indicating a slight cytotoxic effect, whereas extracts at 640, 1280, and 36,000 μg/mL exhibited strong cytotoxicity towards human dermal fibroblasts (cell viability equal to 51%, 21%, and 0% after 48 h exposure time, respectively).

Based on the screening cytotoxicity test, the highest relatively nontoxic or slightly cytotoxic concentrations of common bean sprout extract were selected for further studies, as follows: 40, 80, and 160 μg/mL. The lowest concentrations of the extract, despite having been proven nontoxic towards human dermal fibroblasts (HDFs), were not considered in other experiments because they most likely would not provide any therapeutic effects. The performed WST-8 assay demonstrated that selected concentrations of the *P. vulgaris* sprout extract significantly increased proliferation of human dermal fibroblasts compared to the control cells in a dose-dependent manner (Figure 4A). A similar tendency was observed with respect to collagen synthesis. It was noted that total collagen synthesis by human dermal fibroblasts significantly increased after exposure to the *P. vulgaris* sprout extract in a dose-dependent manner (Figure 4B). Based on all cell culture tests, it may be concluded that the 160 μg/mL *P. vulgaris* sprout extract is a promising candidate to be used as a component of cosmetic formulations to improve the condition and vitality of the skin.

## 3. Materials and Methods

### 3.1. Chemicals and Reagents 

Collagenase from *Clostridium histolyticum*, 2,2-diphenyl-1-picrylhydrazyl radical (DPPH^•^), 2,2′-azino-bis-(3-ethyl-benzothiazole-6-sulfonic acid) (ABTS^●+^), elastase from porcine pancreas, (-)-epigallocatechin gallate (EGCG), Folin–Ciocalteu reagent, N-[3-(2-Furyl)acryloyl]-Leu-Gly-Pro-Ala (FALGPA), N-Succinyl-Ala-Ala-Ala-p-nitroanilide (AAAPVN), and tyrosinase from mushrooms were obtained from Sigma-Aldrich (Steinheim, Germany). 3,4-Dihydroxy-L-phenylalanine (L-DOPA), kojic acid, and epigallocatechin gallate EGCG were from Supelco. Reference substances were from ChromaDex (Irvine, CA, USA). Acetonitrile, formic acid, and water for the LC analysis were from Merck (Darmstadt, Germany). All other chemicals were of analytical grade and were obtained from the Polish Chemical Reagent Company (POCH, Gliwice, Poland). 

### 3.2. Growing Sprouts from Seeds (Phaseolus vulgaris L.)

Bean sprouts were grown in a two-level Tribest TRIFL2000 sprouter with an automatic watering system adapted to the season (spring/summer—every 30 min; autumn/winter—every 60 min). The day before the seeds were transferred to the sprouter, they were soaked in water and left in it for 24 h. After this time, the seeds were transferred to two levels of the sprouting chamber, and the appropriate watering mode was set. The sprouting box was covered with a perforated lid and covered with light-impermeable foil. The temperature in the thermostated room was maintained at 22 °C, and the watering mode was summer (every 30 min). Sprouts were cultivated until the first leaf buds appeared, 5–7 days later. Post-harvest, sprouts were cut off at the seed level and stored at 4 °C in a plastic bag until extraction.

### 3.3. Preparation of the Extract

The obtained common bean sprouts (*Phaseolus vulgaris* L. var. ‘Belmonte’) were crushed into fractions no larger than 5 mm, and then, precisely weighed portions of the raw material were placed in glass bottles with matching caps. For the extraction, 70% ethanol with the addition of 0.4% (*w*/*w*) Brij L35 (Polyoxyethylene (23) lauryl ether) was used.

The raw material was treated with an extractant, and then, the bottles with the raw material and extractant were placed on a shaker and shaken at a speed of 300 rpm for 2 h. Subsequently, the mixture was sonicated at a controlled temperature (40 ± 2 °C) for 20 min [33]. After this period, the extractant was poured off from above the raw material and filtered through Wathman filter paper. The raw material was again treated with the same amount of solvent, and the extraction procedure was repeated under the same conditions. The combined extracts were concentrated under reduced pressure (IKA RV8 Flex evaporator equipped with an IKA HB10 thermostated water bath and an IKA Vacstar control membrane pump) and, after freezing, lyophilized in a vacuum concentrator (Free Zone 1 apparatus; Labconco, Kansas City, KS, USA) to obtain dried residue. Dry extract was weighted and stored in a freezer at −20 °C. The extraction yield was 37.60%.

### 3.4. Total Flavonoid, Phenolic, and Phenolic Acid Content

The total flavonoid (TFC) and total phenolic content (TPC) were determined using colorimetric assays according to previously described methods [33]. Absorbance readings were taken at 430 nm and 680 nm, respectively, using a Pro 200F Elisa Reader (Tecan Group Ltd., Männedorf, Switzerland). TPC was calculated from the calibration curve (R^2^ = 0.9992), employing gallic acid at concentrations ranging from 0.002 to 0.16 mg/mL as a standard. The results were expressed as milligrams of gallic acid equivalent (GAE) per gram of dry extract (DE). TFC was calculated from the calibrated curve (R^2^ = 0.997), utilizing quercetin at concentrations ranging from 0.004 to 0.2 mg/mL as a standard. The findings were expressed as milligrams of quercetin equivalent (QE) per gram of DE. Total phenolic acid (TPAC) content was determined using Arnov’s reagent as outlined in the Polish Pharmacopoeia IX (an official translation of PhEur 7.0) [60]. Absorbance was measured at 490 nm. TPAC was determined from the calibration curve (R^2^ = 0.9745), employing caffeic acid at concentrations ranging from 3.36 to 23.52 μg/mL as a standard. The results were expressed in milligrams of caffeic acid equivalent (CAE) per gram of DE.

### 3.5. LC-ESI-MS/MS Analysis 

The chromatographic measurements were performed using an LC-MS system from Thermo Scientific (Q-EXATCTIVE and ULTIMATE 3000, San Jose, CA, USA) equipped with an ESI source. The ESI was operated in positive polarity modes under the following conditions: spray voltage—4.5 kV; sheath gas—40 arb. units; auxiliary gas—10 arb. units; sweep gas—10 arb. units; and capillary temperature—320 °C. Nitrogen (>99.98%) was employed as the sheath, auxiliary, and sweep gas. The scan cycle used a full-scan event at a resolution of 70,000. A Gemini SYNERGI 4u Polar-RP column (250 × 4.6 mm, 5 μm) and a Phenomenex Security Guard ULTRA LC guard column (both from Phenomenex, Torrance, CA, USA) were employed for chromatographic separation, which was performed using gradient elution. Mobile phase A was 25 mM formic acid in water; mobile phase B was 25 mM formic acid in acetonitrile. The gradient program started at 5% B, increasing to 95% for 60 min; isocratic elution followed (95% B) next for 10 min. The total run time was 70 min at a mobile phase flow rate of 0.5 mL/min. The column temperature was 25 °C. In the course of each run, the MS spectra in the range of 100–1000 *m*/*z* were collected continuously. The amounts of the identified compounds were calculated based on the calibration curves obtained for the standard.

### 3.6. Antioxidant Activity

All experiments were conducted using 96-well microplates (Nunclon, Nunc, Roskilde, Denmark) and analyzed using an Infinite Pro 200F Elisa Reader (Tecan Group Ltd., Männedorf, Switzerland). 

#### 3.6.1. DPPH^•^ Assay

The assessment of the 2,2-diphenyl-1-picryl-hydrazyl (DPPH^•^) free radical scavenging activity of *P. vulgaris* sprout extract was conducted using a modified procedure outlined previously [61]. The decrease in DPPH^•^ absorbance, prompted by the extracts, was monitored at 517 nm following a 30 min incubation period at 28 °C. The results were obtained from individual sample measurements and presented as Trolox equivalents, expressed in milligrams of Trolox per gram of dry extract [mgTE/g DE].

#### 3.6.2. ABTS^•+^ Assay

Another method used to evaluate the antioxidant properties of *P. vulgaris* sprout extract was the 2,2′-azinobis[3-ethylbenzthiazoline]-6-sulfonic acid (ABTS^•+^) decolorization assay [62]. The absorbance was measured after a 6 min incubation at 734 nm. The results were presented as Trolox equivalents, expressed in milligrams of Trolox per gram of dry extract [mgTE/g DE].

### 3.7. Anti-Inflammatory Activity

The inhibitory activity of *P. vulgaris* sprout extract against COX-1 and COX-2 was determined using a COX (ovine/human) Inhibitor Screening Assay Kit (Cayman Chemical, Ann Arbor, MI, USA), following the manufacturer’s protocol. Indomethacin was used as a positive control for inhibiting both COX-1 and COX-2.

### 3.8. Enzyme Inhibitory Activity

#### 3.8.1. Anti-Elastase Activity

The anti-elastase activity was assessed spectrophotometrically following the method outlined by Chiocchio et al. [63], with some modifications. Porcine pancreatic elastase (3.33 mg/mL) and bean sprout samples (5–100 μg/mL) were incubated in Tris-HCl buffer (1.6 mM, pH 8.0) for 15 min at 29 °C. The reaction was initiated by adding N-Succinyl-Ala-Ala-Ala-p-nitroanilide (AAAPVN; 2 mM) as a substrate. The final well reaction mixture in a 96-well plate (total volume 200 µL) contained buffer, 0.8 mM AAAPVN, 1 µg/mL elastase, and 25 µL of extract at the appropriate concentration. After a 20 min incubation period, the absorbance was measured at 410 nm. Epigallocatechin gallate (100 µg/mL) served as the positive control.

#### 3.8.2. Anti-Collagenase Activity

The evaluation of anti-collagenase activity utilized N-[3-(2-furyl)acryloyl]-Leu-Gly-Pro-Ala (FALGPA) as a substrate, following the method described by Mandrone et al. [64]. Collagenase from *Clostridium histolyticum* (20 mU), dissolved in Tricine buffer (50 mM, pH 7.5) at an initial concentration of 0.8 U/mL, and various concentrations of common bean sprout extract were incubated for 15 min at 35 °C. The final reaction mixture (150 μL) contained Tricine buffer, 0.8 mM FALGPA, 0.1 U collagenase, and the appropriate concentration of extract. The absorbance was measured at 345 nm. Epigallocatechin gallate (100 µg/mL) was used as the positive control.

#### 3.8.3. Anti-Tyrosinase Activity

Anti-tyrosinase activity was estimated using the method described earlier by Zengin et al. [65]. For this purpose, different concentrations of the tested extract (20 μL), mushroom tyrosinase solution (200 U/mL, 40 μL), and phosphate buffer (100 μL, 50 mM, pH 6.5) were applied to a 96-well plate and then incubated for 15 min at 25 °C. The reaction was started with the addition of L-DOPA (L-3,4-dihydroxyphenylalanine; 40 µL, 0.5 mM) and incubated for another 15 min, after which the absorbance was measured at 475 nm compared to the control without the inhibitor. In the control sample, representing 100% of the enzyme activity, phosphate buffer was added instead of the extract. Tyrosinase inhibitory activity was determined by comparing enzyme activity in samples with and without the inhibitor being evaluated. Kojic acid was used as a standard at a concentration of 100 µL/mL.

### 3.9. Cell Culture Experiments

The screening cytotoxicity test, cell proliferation assay, and collagen synthesis tests were carried out using primary human dermal fibroblasts (HDFs, ATCC-LGC Standards). Cells were cultured at 37 °C in a humidified atmosphere of 5% CO_2_ and 95% air. Fibroblasts were maintained in a Fibroblast Basal Medium supplemented with the components of Fibroblast Growth Kit-Low Serum (both purchased from ATCC-LGC Standards) and antibiotics (100 U/mL penicillin and 0.1 mg/mL streptomycin purchased from Sigma-Aldrich Chemicals, Warsaw, Poland).

### 3.10. Screening Cytotoxicity Test

Different concentrations of the bean extract, ranging between 5 and 3600 μg/mL, were prepared by dissolving the lyophilized sample in the complete culture medium, followed by serial two-fold dilutions. To test cytotoxicity, an HDF suspension with a concentration of 2 × 10^5^ cells/mL was seeded in 100 μL into the wells of 96-multiwell plates. After 24 h incubation at 37 °C, the culture medium was discarded, and the monolayer of cells was exposed to 100 µL of the different concentrations of bean sprout extract. HDFs maintained in the culture medium without bean sprout extract served as a control. Cells were cultured for 24 h and 48 h, and then cytotoxicity was determined by evaluation of cell metabolism using the MTT (3-(4,5-dimethylthiazol-2-yl)-2,5-diphenyltetrazolium bromide) assay (Sigma-Aldrich Chemicals, Warsaw, Poland). For this purpose, 100 μL of freshly prepared MTT solution (5 mg/mL) in buffered saline (PBS) was added to the wells. After 3 h of incubation at 37 °C, 100 µL of 10% sodium dodecyl sulfate (SDS) solution (Sigma-Aldrich Chemicals, Warsaw, Poland) in 0.01 M HCl (Sigma-Aldrich Chemicals, Warsaw, Poland) was added. The 96-multiwell plates were placed in the incubator at 37 °C for 8 h, and then the absorbance value was measured at a wavelength of 570 nm using a microplate reader (BioTek Synergy H4 Hybrid Microplate Reader, Winooski, VT, USA). The results obtained from the colorimetric test were presented as a percentage of the absorbance value obtained with the control cells, showing 100% viability. Based on these cytotoxicity tests, the highest concentrations of bean sprout extract that did not reduce cell viability below 80% were selected for further cell experiments.

### 3.11. Cell Proliferation Assessment

HDFs were seeded in 96-multiwell plates in 100 μL of complete culture medium at a concentration of 1 × 10^5^ cells/mL per well. After 24 h of incubation at 37 °C, the culture medium was replaced with 100 μL of the selected concentrations of bean sprout extracts (40, 80, and 160 µg/mL). On the third day of the experiment, half of the extract was replaced with a new portion. After 1-, 2-, and 5-day exposure of cells to the extracts, cell number was calculated using Cell counting kit-8 (WST-8 assay, Sigma-Aldrich Chemicals, Warsaw, Poland) following the manufacturer’s instructions.

### 3.12. Evaluation of Collagen Synthesis

A suspension of human dermal fibroblasts (HDFs) at a concentration of 1 × 10^5^ cells/mL was seeded in 100 μL into the wells of 96-multiwell plates. After 24 h of incubation at 37 °C, the culture medium was replaced with 100 μL of the appropriate bean sprout extract prepared in the culture medium. On the third day of the experiment, half of the extract was replaced with a new portion. After 5-day culture in the presence of bean sprout extracts, collagen synthesis was assessed by the colorimetric Sinus Red Total Collagen Detection Assay Plate Kit (Chondrex, Woodinville, WA, USA). The test was performed in accordance with the manufacturer’s protocol.

### 3.13. Statistical Analysis

All experiments were conducted in three independent tests. For the cytotoxicity screening test, statistically significant differences between control cells maintained in the culture medium and various concentrations of the extract were considered at *p* < 0.05 according to a One-way ANOVA with post hoc Dunnett’s test (GraphPad Prism 8.0.0 Software, San Diego, CA, USA). In the case of the proliferation assay and collagen synthesis evaluation, statistically significant results between all tested groups were considered at *p* < 0.05 according to One-way ANOVA with post hoc Tukey’s test.

## 4. Conclusions

In summary, our findings suggest that the micellar extract of *P. vulgaris* sprouts serves as a significant reservoir of natural phenolic compounds, potentially beneficial in mitigating symptoms associated with oxidative stress, such as skin aging. This aligns with previous research on various foods [37,66], which highlighted polyphenolic compounds, including quercetin derivatives, as key contributors to the observed antioxidant effects. In our in vitro study, we characterized the composition and evaluated the biological activities of micellar extract from the sprouts of *P. vulgaris*. Thus, we identified the main compounds present in the extracts and determined the cytotoxic, anti-inflammatory, antioxidant, anti-collagenase, anti-elastase, and anti-tyrosinase properties, as well as the collagen synthesis capabilities of this extract. Our findings indicate that the extract is rich in caffeic acid, ferulic acid, rutin, hyperoside, and quercitrin, which are likely responsible for its beneficial biological activities.

Moreover, the use of common bean sprout extract in micellar form allows for a stronger effect in each test compared to other extracts. Similar conclusions were reached by Ullah A et al. [67], who indicated that new methods for introducing active substances into drug formulations (e.g., liposomes, nanoparticles, and micelles) allow for more effective action of the preparations and expand their pharmacokinetic profiles. The authors indicate that sanguinarine, isolated from *Sanguinaria canadensis* roots and *Argemone mexicana* seeds, has a stronger anti-inflammatory effect when introduced into a modern carrier—solid lipid nanoparticles—than when administered without a carrier. Research obtained by Ullah A. [68] also proved that sanguinarine can be an effective anticancer agent by influencing ROS.

The results of our study revealed that the micellar extract of *P. vulgaris* sprouts could be used as a readily accessible source of cosmetic ingredients. The extract from bean sprouts can be incorporated into various cosmetic products, such as creams, serums, masks, and eye care products. Thanks to its anti-wrinkle and antioxidant properties, this extract can significantly improve skin elasticity, reduce the appearance of wrinkles, and support epidermal regeneration. Despite its numerous advantages, there are some limitations to using bean sprout extract. These include the possibility of allergic reactions in individuals sensitive to bean components. Additionally, the stability and effectiveness of the extract may depend on its storage method and formulation in cosmetic products. Another important aspect is the lack of long-term clinical studies that confirm the safety and effectiveness of the extract for daily use. Therefore, future research should focus on conducting long-term clinical studies to confirm the safety and efficacy of the extract in various cosmetic products and developing cosmetic formulations that will better utilize the properties of bean sprout extract (*Phaseolus vulgaris* L.), ensuring its optimal bioavailability and effectiveness. It is also essential to identify potential allergens in the extract and develop methods to minimize the risk of allergic reactions.

## Figures and Tables

**Figure 1 molecules-29-03058-f001:**
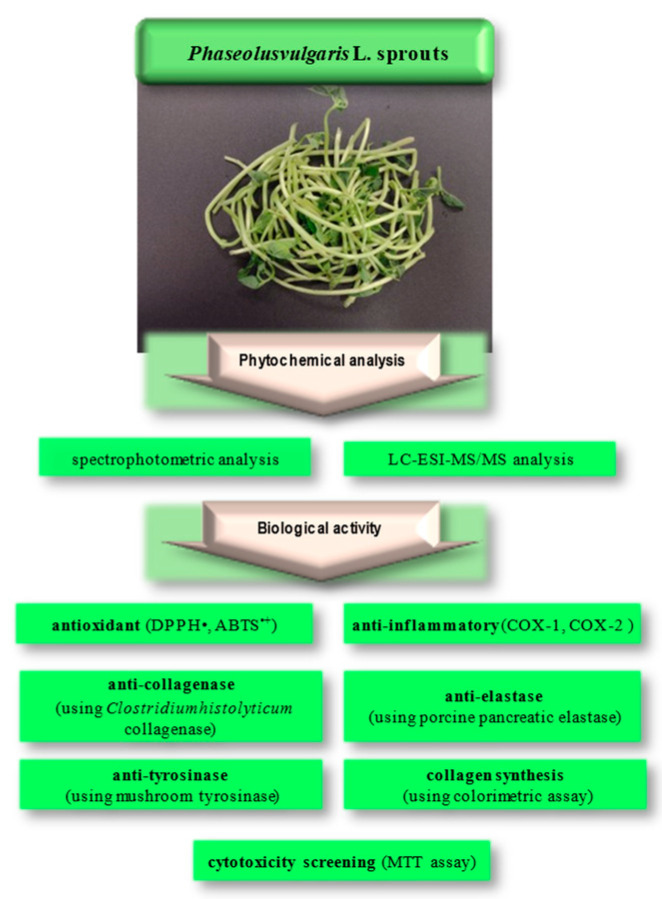
Plan of current research on *Phaseolus vulgaris* sprouts.

**Figure 2 molecules-29-03058-f002:**
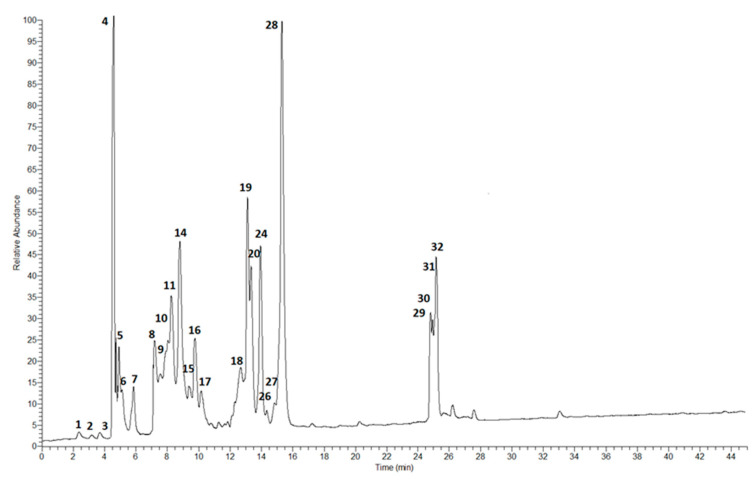
LC-MS chromatogram in SCAN mode for ethanol extract of *P. vulgaris* sprout extract.

**Figure 3 molecules-29-03058-f003:**
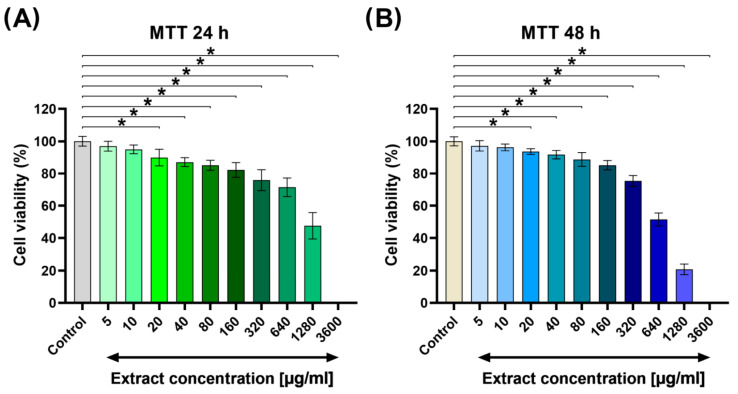
Screening cytotoxicity test on common bean sprout extract towards human dermal fibroblasts: (**A**) (3-(4, 5-dimethylthiazolyl-2)-2, 5-diphenyltetrazolium bromide) (MTT) assay performed after 24 h and (**B**) 48 h exposure time to the extracts (Control—cells maintained in the culture medium without common bean sprout extract; * statistically significant results considered at *p* < 0.05 compared to the control cells according to One-way ANOVA with post hoc Dunnett’s test).

**Figure 4 molecules-29-03058-f004:**
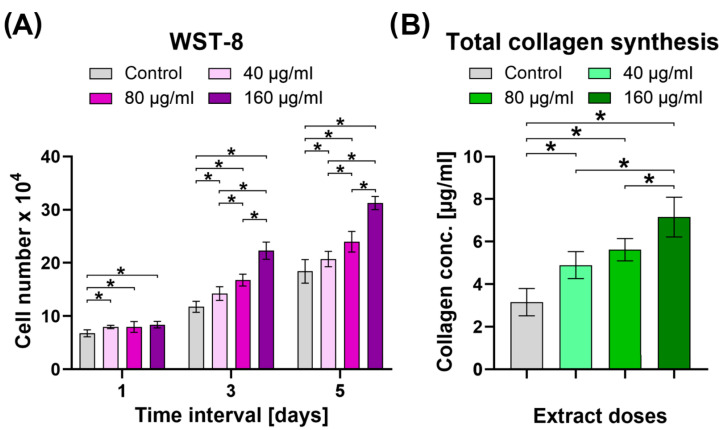
Biological characterization of selected concentrations of common bean sprout extract. (**A**) Proliferation test using human dermal fibroblasts; (**B**) total collagen synthesis by human dermal fibroblasts after 5-day exposure to the extract (Control—cells maintained in the culture medium without common sprout extract; * statistically significant results considered at *p* < 0.05 between indicated groups according to One-way ANOVA with post hoc Tukey’s test).

**Table 1 molecules-29-03058-t001:** Total phenolic (TPC), flavonoid (TFC), and phenolic acid (TPAC) content in the micellar extract of *P. vulgaris* sprouts (PVS); DE—dry extract; GAE—gallic acid equivalent; CAE—caffeic acid equivalent; QE—quercetin equivalent. Values are presented as mean ± standard deviation (*n =* 9).

Sample	Total Phenolic Content [mg GAE/g DE]	Total Phenolic Acids [mg CAE/g DE]	Total Flavonoid Content [mg QE/g DE]
PVS	192.85 ± 10.24	4.66 ± 0.29	178.73 ± 1.59

**Table 2 molecules-29-03058-t002:** High-resolution mass spectrometry (HR-MS) of [M-H]^-^ ion and MS^2^ data. b—base peak.

Peak No.	Compound Name	[M-H]^−^	MS^2^	Theoretical Mass [M-H]^−^ (Da)	Experimental Mass [M-H]^−^ (Da)	Δ mDa	Δ ppm	Elemental Composition
1	Gallic acid	169	69, 79, 81, 97, 124, 125b	169.01370	169.01373	0.03	0.18	C_7_H_6_O_5_
2	Chlorogenic acid	353	179, 191b	353.08726	353.08733	0.07	0.20	C_16_H_18_O_9_
3	*p*-Hydroxybenzoic acid	137	92, 93, 136, 137b	137.02387	137.02385	−0.02	0.15	C_7_H_6_O_3_
4	*p*-Coumaric aldaric acid	355	119, 163b, 311	355.06653	355.06650	−0.03	0.08	C_15_H_16_O_10_
5	*o*-Coumaric aldaric acid	355	119, 163b, 311	355.06653	355.06651	−0.02	0.06	C_15_H_16_O_10_
6	Protocatechuic acid	153	108, 109, 153b	153.01879	153.01874	−0.05	0.33	C_7_H_6_O_4_
7	Feruroyl aldaric acid	385	73, 89, 106, 117, 133, 134, 179, 193b, 341	385.07709	385.07712	0.03	0.08	C_16_H_18_O_11_
8	Caffeic acid	179	97, 107, 135b	179.03444	179.03447	0.03	0.17	C_9_H_8_O_4_
9	*p*-Coumaric acid	163	119, 162b	163.03952	163.03949	−0.03	0.18	C_9_H_8_O_3_
10	*o*-Coumaric acid	163	119, 162b	163.03952	163.03951	−0.01	0.06	C_9_H_8_O_3_
11	Ferulic acid	193	73, 89b, 106, 117, 133, 134, 179	193.05009	193.05015	0.06	0.31	C_10_H_10_O_4_
12	Vanillic acid	178	108, 123, 152, 167b	178.04774	178.04768	0.09	0.54	C_8_H_8_O_4_
13	Quercetin-3-*O*-xylosylglucoside	595	151, 179, 192, 209, 301b, 421	595.12992	595.12995	0.04	0.06	C_26_H_28_O_16_
14	Rutin	609	151, 179, 301b, 463, 609	609.18195	609.18192	−0.03	0.05	C_27_H_30_O_16_
15	Quercetin-3-*O*-(6″-*O*-malonyl) glucoside	549	151, 179, 192, 209, 301b, 375	549.08805	549.08809	0.04	0.07	C_24_H_22_O_15_
16	Kaempferol-3-*O*-glucosylxylose	579	93, 97, 119, 164, 285b, 435	579.13500	579.13507	0.07	0.12	C_26_H_28_O_15_
17	Kaempferol-3-*O*-(6″-*O*-malonyl) glucoside	533	93, 97, 119, 164, 285b, 359	533.09314	533.09311	−0.02	0.05	C_24_H_22_O_14_
18	Hyperoside	463	151, 179, 255, 271, 301b	463.08766	463.08769	0.03	0.06	C_21_H_20_O_12_
19	Isoquercitrin	463	151, 179, 192, 301, 461, 463b	463.08766	463.08762	−0.04	0.09	C_21_H_20_O_12_
20	Quercitrin	447	151, 243, 255, 271, 300b, 301, 447	447.09274	447.09278	0.04	0.09	C_21_H_20_O_11_
21	Sinapic aldaric acid	415	141, 149, 164, 208, 223b, 371	415.08766	415.08761	−0.05	0.11	C_17_H_20_O_12_
22	Myricetin-3-*O*-glucoside	479	137, 153, 165, 229b, 257	479.08257	479.08252	−0.05	0.10	C_21_H_20_O_13_
23	Prunin	433	119, 151, 271b	433.11348	433.11353	0.06	0.13	C_21_H_22_O_10_
24	Astragalin	447	257, 285b, 447	447.09274	447.09281	0.07	0.16	C_21_H_20_O_11_
25	Myricetin	317	107, 109, 137, 151, 179, 317b	317.02975	317.02970	−0.05	0.16	C_15_H_10_O_8_
26	Sinapic acid	223	141, 149, 164b, 208, 223	223.06065	223.06068	0.03	0.13	C_11_H_12_O_5_
27	Quercetin	301	151, 179b, 192, 209	301.03483	301.03481	−0.02	0.07	C_15_H_10_O_7_
28	(+)-Catechin	289	109, 123, 125, 137, 165, 179, 203, 205, 245, 289b	289.07122	289.07129	0.07	0.24	C_15_H_14_O_6_
29	Kaempferol	285	93, 97, 119b, 164, 285	285.03992	285.03980	−0.12	0.42	C_15_H_10_O_6_
30	Naringenin	271	107, 119, 151b, 177, 271	271.06065	271.06071	0.06	0.22	C_15_H_12_O_5_
31	Daidzein	253	89, 117, 135b, 151, 169, 179, 227	253.05009	253.05013	0.05	0.18	C_15_H_10_O_4_
32	Glycitein	283	240, 268b, 269, 283	283.06065	283.06060	−0.05	0.18	C_16_H_12_O_5_

**Table 3 molecules-29-03058-t003:** Content of active compounds in *Phaseolus vulgaris* sprout micellar extract; DE—dry extract.

Peak No.	Compound Name	Calibration Standards	Amounts [μg/g DE]
1	Gallic acid	Gallic acid	25.3 ± 1.3
2	Chlorogenic acid	Chlorogenic acid	467.8 ± 20.9
3	*p*-Hydroxybenzoic acid	*p*-Hydroxybenzoic acid	4693.5 ± 214.0
4	*p*-Coumaryl aldaric acid	*p*-Coumaric acid	58,668.2 ± 3056.6
5	*o*-Coumaryl aldaric acid	*p*-Coumaric acid	21,923.4 ± 1091.8
6	Protocatechuic acid	Protocatechuic acid	109.9 ± 5.2
7	Feruroyl aldaric acid	Feruroyl acid	10,266.9 ± 541.1
8	Caffeic acid	Caffeic acid	11,857.2 ± 550.2
9	*p*-Coumaric acid	*p*-Coumaric acid	1605.7 ± 85.3
10	*o*-Coumaric acid	*p*-Coumaric acid	1442.0 ± 65.9
11	Ferulic acid	Ferulic acid	11,208.7 ± 558.2
12	Vanillic acid	Vanillic acid	37.2 ± 1.9
13	Quercetin-3-*O*-xylosylglucoside	Rutin	1204.2 ± 60.5
14	Rutin	Rutin	10,637.5 ± 529.7
15	Quercetin-3-*O*-(6″-*O*-malonyl) glucoside	Rutin	1151.7 ± 52.6
16	Kaempferol-3-*O*-glucosylxylose	Rutin	2640.1 ± 133.6
17	Kaempferol-3-*O*-(6″-*O*-malonyl) glucoside	Rutin	1347.8 ± 59.8
18	Hyperoside	Rutin	3597.3 ± 173.4
19	Isoquercitrin	Rutin	290.3 ± 14.7
20	Quercitrin	Rutin	2037.9 ± 106.2
21	Sinapyl aldaric acid	Sinapic acid	378.3 ± 17.1
22	Myricetin-3-*O*-glucoside	Rutin	217.7 ± 10.1
23	Prunin	Rutin	140.2 ± 7.1
24	Astragalin	Astragalin	3504.7 ± 174.5
25	Myricetin	Myricetin	54.7 ± 2.8
26	Sinapic acid	Sinapic acid	1442.0 ± 74.6
27	Quercetin	Quercetin	683.9 ± 32.0
28	(+)-Catechin	(+)-Catechin	41,530.9 ± 2126.4
29	Kaempferol	Kaempferol	2809.9 ± 128.4
30	Naringenin	Kaempferol	1215.0 ± 55.3
31	Daidzein	Kaempferol	1165.6 ± 59.3
32	Glycitein	Kaempferol	1133.2 ± 54.1

**Table 4 molecules-29-03058-t004:** Antioxidant activity of *P. vulgaris* sprout extract. Antioxidant activity expressed as the DPPH^•^ scavenging assay [expressed as mg Trolox (Trolox equivalents)/g of dry extract] and antiradical capacity (ABTS^+•^) [expressed as mg Trolox (Trolox equivalents)/g of dry extract]. Values are presented as the mean ± standard deviation of triplicate measurements.

	DPPH [mgTE/g]	ABTS [mgTE/g]
Micellar extract	246.24 ± 0.05	74.38 ± 0.01

**Table 5 molecules-29-03058-t005:** Inhibition of COX-1 and COX-2 cyclooxygenase activity by *P. vulgaris* sprout extract.

Micellar Extract [μg/mL]	COX-1 Inhibition [%] ± SD	COX-2 Inhibition [%] ± SD
10	6.90 ± 0.14	6.25 ± 0.42
25	16.79 ± 0.44	11.31 ± 0.25
50	60.80 ± 0.28	52.57 ± 0.43
100	57.22 ± 0.31	48.86 ± 0.35
Indomethacin	61.15 ± 0.26	55.29 ± 0.17

**Table 6 molecules-29-03058-t006:** Anti-collagenase, anti-elastase, and anti-tyrosinase activities of *P. vulgaris* sprout extract; nt—not tested; EGCG—epigallocatechin gallate.

Micellar Extract [μg/mL]	Collagenase Inhibition	Elastase Inhibition	Tyrosinase Inhibition
5	8.40 ± 0.41	10.62 ± 0.36	18.35 ± 0.43
10	14.22 ± 0.17	18.10 ± 0.17	26.13 ± 0.12
25	31.72 ± 0.29	42.71 ± 0.42	35.70 ± 0.54
50	69.69 ± 0.38	67.18 ± 0.30	49.84 ± 0.26
100	61.27 ± 0.15	86.29 ± 0.35	58.33 ± 0.46
EGCG	83.29 ± 0.36	90.35 ± 0.43	nt
Kojic acid	nt	nt	65.49 ± 0.25

## Data Availability

The data presented in this study are available upon request from the corresponding authors due to privacy.

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
