# Peer review of "Novel Insights into Phaseolus vulgaris L. Sprouts: Phytochemical Analysis and Anti-Aging Properties"

_molecules, 2024, doi:10.3390/molecules29133058_

Round 1
Reviewer 1 Report
Comments and Suggestions for Authors
The manuscript titled "Novel Insights Into Phaseolus Vulgaris L. Sprouts - Phytochemical Analysis and Anti-aging Properties" presents an in-depth study of the phytochemical composition and anti-aging effects of Phaseolus vulgaris L. sprouts. The research highlights the identification and quantification of bioactive compounds and evaluates their potential in mitigating age-related cellular damage. These findings offer valuable contributions to the understanding of the health benefits associated with Phaseolus vulgaris L. sprouts and suggest their applicability in anti-aging therapies. The manuscript can be recommended for publication after major changes and amendments.
1. The authors should discuss these relevant references in the introduction, and the discussion which will strengthen the manuscript https://doi.org/10.2174/1871520622666220831124321, https://doi.org/10.2174/0118761429269383231119062233
2. The authors should add a graphical or schematic diagram in the last part of the introduction
3. Line 451, which season the bean sprouts were grown?
4. Line 464, ethanol 70°? Is it 70% ethanol?
5. Line 301, the authors should correct this [37.40].
6. Acronyms and abbreviations need to be defined when they are first used.
7. The authors need to recheck the whole manuscript to remove grammatical errors/ typos/ incomplete sentences and non-relative phrases.
Reviewer 2 Report
Comments and Suggestions for Authors
Dear Authors,
I have carefully reviewed your manuscript titled "Novel Insights Into Phaseolus Vulgaris L. Sprouts - Phytochemical Analysis and Anti-aging Properties," and I am pleased to provide you with the following review report.
Overall, your manuscript presents a comprehensive and well-designed study, evaluating the potential of Phaseolus vulgaris sprout extract for anti-aging applications. The introduction provides a clear rationale for the study, highlighting the need for natural and safe alternatives in cosmetic formulations. The objectives and methodologies are well-described, and the results are thoroughly presented and discussed.
One of the strengths of your study is the detailed phytochemical analysis of the extract using LC-MS. The identification and quantification of various phenolic compounds, including phenolic acids, flavonoids, and their derivatives, provide valuable insights into the bioactive components present in the extract. The comparison of your findings with previous studies on common bean and other sprout extracts is commendable and helps to contextualize the results.
The evaluation of the extract's biological activities, such as antioxidant, anti-inflammatory, anti-collagenase, anti-elastase, and anti-tyrosinase properties, is another notable aspect of your study. The methods used for these assays are well-established and appropriately described. The results demonstrate the promising potential of the extract in exhibiting anti-aging properties, which is further supported by the comparison with relevant literature findings.
Furthermore, the cytotoxicity and cell culture studies add significant value to your work. The determination of safe and effective concentrations of the extract, as well as the evaluation of its effect on cell proliferation and collagen synthesis, provide crucial information for potential applications in cosmetic formulations. The observed promotion of cell proliferation and collagen synthesis without significant cytotoxicity is an encouraging finding.
However, there are a few minor points that could be addressed to further strengthen your manuscript:
1. In the introduction, it would be beneficial to provide a more detailed overview of the existing knowledge and gaps related to the anti-aging properties of Phaseolus vulgaris sprouts or similar sprout extracts.
2. In the discussion section, you could elaborate further on the potential mechanisms of action underlying the observed biological activities, considering the identified phytochemical components.
3. While the manuscript focuses on the anti-aging properties of the extract, it would be valuable to discuss the potential applications and limitations of the extract in cosmetic formulations, as well as any future perspectives or recommendations for further research.
Overall, your manuscript presents a well-executed study with significant contributions to the field of natural anti-aging compounds. The findings provide a strong foundation for further exploration and potential development of cosmetic formulations incorporating Phaseolus vulgaris sprout extract.
I hope this review report is helpful, and I look forward to seeing your manuscript published after addressing the minor points mentioned above.
Best regards
Comments on the Quality of English Language
Extensive editing of English language required
Round 2
Reviewer 1 Report
Comments and Suggestions for Authors
The manuscript titled "Novel Insights Into Phaseolus Vulgaris L. Sprouts - Phytochemical Analysis and Anti-aging Properties" presents an in-depth study of the phytochemical composition and anti-aging effects of Phaseolus vulgaris L. sprouts. The research highlights the identification and quantification of bioactive compounds and evaluates their potential in mitigating age-related cellular damage. These findings offer valuable contributions to the understanding of the health benefits associated with Phaseolus vulgaris L. sprouts and suggest their applicability in anti-aging therapies. The author has addressed some concerns, but not all the concerns, further improvement is necessary before recommending this manuscript for publication, particularly through implementing major changes.
1. Line 17 -18, Skin aging is an inevitable, and intricate process instigated by oxidative stress, culminating in heightened activation of extracellular matrix-disrupting enzymes and DNA damage. This sentence is confusing, please modify the language.
2. Line 20, please add the methods in the abstract.
3. I mentioned in my previous comments, that the authors should add a graphical abstract or sketch in the last part of the introduction.
4. Line 404, add vendor name, Supelco….here, also Reference substances include name.
5. Line, add the experimental condition
6. Line 263, add the Table for section 2.2
7. Line 302, Add the control in Table 4 and the same for Table 5.
8. Line 345, add the Table for section 2.5
9. Line 39 – 40, many species' efficacy still 39 hinges on anecdotal evidence rather than robust scientific validation facilitated by modern analytical methodologies. Please modify it, it's not clear.
10. The author didn't add the relevant references as mentioned in my previous comments.
11. There is too much jargon in the article, please reduce it.
12. The authors need to double-check the whole manuscript to remove grammatical errors/ typos/incomplete sentences and non-relative phrases.
Comments on the Quality of English LanguageThere are too many jargon and grammatical mistakes in the whole manuscript. Please reduce the jargon and improve the English language.
Reviewer 2 Report
Comments and Suggestions for Authors
Accept in the revised form
Comments on the Quality of English Language
Extensive editing of English language required
Author Response
Dear Reviewer,
Thank you for your positive opinion of our manuscript.
Sincerely Yours,
Authors